# RELIGHTMASTER: PRECISE VIDEO RELIGHTING WITH MULTI-PLANE LIGHT IMAGES

## ABSTRACT

Recent advances in diffusion models enable high-quality video generation and editing, but precise relighting with consistent video contents, which is critical for shaping scene atmosphere and viewer attention, remains unexplored. Mainstream text-to-video (T2V) models lack fine-grained lighting control due to text's inherent limitation in describing lighting details and insufficient pre-training on lighting-related prompts. Additionally, constructing high-quality relighting training data is challenging, as real-world controllable lighting data is scarce. To address these issues, we propose RelightMaster, a novel framework for accurate and controllable video relighting. First, we build RelightVideo, the first dataset with identical dynamic content under varying precise lighting conditions based on the Unreal Engine. Then, we introduce Multi-plane Light Image (MPLI), a novel visual prompt inspired by Multi-Plane Image (MPI). MPLI models lighting via $K$ depth-aligned planes, representing 3D light source positions, intensities, and colors while supporting multi-source scenarios and generalizing to unseen light setups. Third, we design a Light Image Adapter that seamlessly injects MPLI into pre-trained Video Diffusion Transformers (DiT): it compresses MPLI via a pre-trained Video VAE and injects latent light features into DiT blocks, leveraging the base model's generative prior without catastrophic forgetting. Experiments show that RelightMaster generates physically plausible lighting and shadows and preserves original scene content.

## 1 INTRODUCTION

With the advancement of video generation technology (Blattmann et al., 2023; Wan et al., 2025), it is now possible to generate high-quality, long video clips comparable to movies. Improving the controllability of video generation is a pressing need for downstream applications, e.g., camera trajectory control (Bian et al., 2025; He et al., 2025) and multi-identity preservation (Liu et al., 2025a). As a fundamental element of video content creation, lighting plays an irreplaceable role: it shapes the visual atmosphere of scenes, enhances spatial depth, and guides viewers' attention—directly determining the aesthetic and communicative effect of video content. However, achieving precise lighting control and flexible lighting editing remains highly challenging. Traditionally, professional video lighting relied on specialized equipment or specific environmental conditions, which are difficult for ordinary creators to replicate. Even with the lowered creation threshold brought by video generation models, mainstream text-to-video (T2V) models still fail to support accurate, fine-grained lighting control, creating a critical gap between technical capability and practical demand. We propose a framework RelightMaster for precise video relighting.

We observe that there are two core challenges hindering the progress of video relighting. First, constructing high-quality training data for relighting is extremely difficult. Real-world video data with controllable lighting conditions is scarce: adjusting lighting parameters in physical scenes is time-consuming, costly, and unable to ensure consistent content across different lighting setups. To mitigate this, we turn to game engines (e.g., Unreal Engine 5 (Games, 2022)) to generate synthetic data. Nevertheless, such synthetic data still has limitations: its appearance (e.g., texture details, color realism) differs significantly from general real-world video data, and the limited data volume easily leads to model overfitting. This calls for an effective method to activate the prior knowledge already learned by pre-trained video generation models, bridging the gap between synthetic and real data. Second, representing and inputting lighting information accurately is a bottleneck. T2V

models primarily take text prompts as input, but text is inherently inadequate for describing fine-grained lighting details (e.g., light position, intensity distribution, color temperature). Worse still, the prompts used in T2V pre-training rarely include lighting-related descriptions, leaving models unable to learn effective lighting representations from text, which further limits the precision of lighting control. To address this challenge, we argue that a visual prompt is needed: one that can not only provide precise, quantitative control signals for light sources (e.g., positional and intensity information) but also naturally align with the video prior (i.e., the visual distribution and spatial structure learned by pre-trained video generation models), thus overcoming the inaccuracy of text-based lighting control while leveraging existing model knowledge.

We draw inspiration from the multi-plane image (MPI) (Tucker & Snavely, 2020) representation and propose a novel Multi-plane Light Image (MPLI) for video relighting. The core idea of MPLI is to model lighting information in a spatially aligned manner with video content: we first extract $K$ depth planes from the camera frustum, covering the spatial hierarchy of the scene. Then, we calculate the irradiance on each of these $K$ planes based on the 3D position of the light source, generating $K$ corresponding Light Images. This design endows MPLI with three key advantages: (1) it fully captures the 3D positional information of light sources, establishing a natural alignment with the 2D frame modality of video; (2) it inherently supports multi-light-source scenarios. Multiple light sources can be integrated by superimposing their respective irradiance calculations on the $K$ planes; (3) it exhibits strong generalization: in our experiments, even when trained only on single-light-source data, the model naturally supports multi-light-source relighting, verifying the robustness of the MPLI.

To seamlessly integrate MPLI as a control condition into existing video generation models, we further propose a Light Image Adapter. Current video generation models typically use a Video Variational Autoencoder (VAE) to compress $K$ video frames into a single video latent feature, which is then processed by a patchify module and fed into a Diffusion Transformer (DiT) (Peebles & Xie, 2023) for generation. To align MPLI with the visual distribution learned by the diffusion model, we first compress the $K$ Light Images of MPLI into a single latent light feature using the same pre-trained Video VAE. The Light Image Adapter is initialized with parameters from the pre-trained patchify module, which ensures consistency with the model's prior knowledge, and injects the latent light feature into the network before each DiT block. This lightweight integration not only preserves the original generation capability of the DiT model but also enables precise, fine-grained control over video relighting. In contrast, previous methods (Zhou et al., 2025; Zhang et al., 2024) only relit videos with rough texts or replace the background with environment maps.

The main contributions of our work can be summarized as follows:

- We propose a novel light representation, the Multi-plane Light Image (MPLI), that explicitly encodes the spatial properties of 3D light sources and aligns naturally with the video modality. The MPLI enables dynamic multisource light control and demonstrates strong generalization.
- We propose a lightweight and efficient Light Image Adapter that seamlessly injects the MPLI condition into pre-trained Video DiT models. This allows for precise lighting control while leveraging the vast generative prior of the base model, avoiding catastrophic forgetting and the need for full retraining.
- We build a dataset, RelightVideo, the first video dataset that renders the same dynamic contents with different lighting conditions, advancing the cutting-edge research on light control in video generation and editing.
- We propose a novel framework, RelightMaster, for accurate and controllable video relighting that generates physically plausible lighting and shadow effects across the entire scene while preserving the original background content.

## 2 RELATED WORKS

**Diffusion Models for Relighting**. Traditional image relighting methods rely on intrinsic image decomposition (Luo et al., 2020; Careaga & Aksoy, 2023; Liang et al., 2025), which decomposes an sRGB image into shading, albedo, and then replaces the shading according to the estimated normals. Recently, text-to-image (T2I) diffusion models (Dhariwal & Nichol, 2021; Ho et al., 2020;

Rombach et al., 2022; Song et al., 2020) have emerged as pivotal foundational models in image editing, attributed to their strong capability in learning real-world image priors. For the task of image relighting, a prominent approach is fine-tuning these pre-trained T2I models. Such methods eliminate the need for explicit decomposition of intrinsic scene components (e.g., shape, albedo) and directly leverage the learned priors of lighting and scene consistency to achieve flexible and realistic illumination editing, supporting diverse control modalities like text descriptions and environment maps. Representative works include LightIt (Kocsis et al., 2024), DiLightNet (Zeng et al., 2024), IC-Light (Zhang et al., 2024), and LightLab (Magar et al., 2025) . Recently, Light-A-Video (Zhou et al., 2025), TC-Light (Liu et al., 2025b), and RelightVid (Fang et al., 2025) extend the image relighting method IC-Light to video relighting. Lumen (Zeng et al., 2025) replaces the background in videos while correspondingly adjusting the lighting in the foreground with harmonious blending.

**Diffusion Models for Video Editing**. Diffusion-based video generation techniques have also undergone remarkable advancements (Blattmann et al., 2023; Wan et al., 2025; Singer et al., 2025; Ling et al., 2024). Leveraging these developments, training-free paradigms including AnyV2V (Ku et al., 2024), MotionClone (Ling et al., 2024), and BroadWay (Bu et al., 2024) facilitate prompt-guided operations such as inpainting, style transfer, and motion retargeting without requiring additional model fine-tuning. For achieving frame-level consistency in edited content, fine-tuning-based approaches like ConsistentVideoTune (Cheng et al., 2023) and Tune-A-Video (Wu et al., 2022) adapt pre-trained video diffusion models to user-provided references, enabling seamless object insertion and consistent color grading effects. Recently, a series of video relighting methods (Zhou et al., 2025; Liu et al., 2025b; Fang et al., 2025) extend IC-Light (Zhang et al., 2024) from image relighting to video relighting. All of the three methods inherent the nature of IC-Light: most lighting controllability comes from the environment map instead of the input text. However, once using the environment map, it requires handcrafting the foreground objects for relighting and enforces static background replacement based on provided environment maps, which does not meet the requirements of video relighting under general circumstances. Camera trajectory editing (Bian et al., 2025; He et al., 2025; Gu et al., 2025; Bai et al., 2025) can be regarded as a type of video editing. Inspired by ReCamMaster (Bai et al., 2025) that synthesized video pairs that share the same dynamic contents via graphics engines, we propose RelightMaster that learns relighting from rendered video pairs. In contrast to previous video relighting methods (Zhou et al., 2025; Liu et al., 2025b; Fang et al., 2025), our proposed RelightMaster achieves good light control with end-to-end generation while preserving the complete original video content.

## 3 DATASET

Collecting video pairs with varying lighting conditions in the real world is challenging. Setting up lighting in real scenes is time-consuming and expensive, which limits the data diversity and scalability. For example, light stages commonly used to collect 3D human body data often feature monotonous backgrounds. Even worse, it is difficult to ensure that the dynamic objects remain consistent across multiple video recordings. Using synthetic data can effectively circumvent the problem of inconsistent motion, and advanced game engines can provide extremely realistic lighting simulations at a low cost.

We build a dataset rendering pipeline based on Unreal Engine to batch generate video training data with the same content but different lighting. We collected 24 3D scene assets as static backgrounds and randomly bound 93 actions to 66 human models as dynamic object foregrounds. Finally, we obtained 652 assembled scenes after random combination. Fig. 1 presents an overview of our dataset. For each scene, we use four random camera trajectories centered on dynamic objects to render the original video, that is, the reference video without changing the lighting conditions. We then add additional point lights with randomized parameters to the existing scene and render the target video again with the changed lighting conditions. We adjust the 3D position, color, and intensity of the point lights to provide fine-grained control over the lighting conditions. We focus on the main parameters that determine the basic physical properties of light sources. The coordinates of the light source are always relative to the first frame of the video, with the camera center as the origin, and do not change as the camera moves. Except for the controllable parameters, all other intrinsic parameters of the light source provided by Unreal Engine 5 are completely fixed. A fixed light source refers to a light source whose parameters are always fixed during the video recording, while a variable

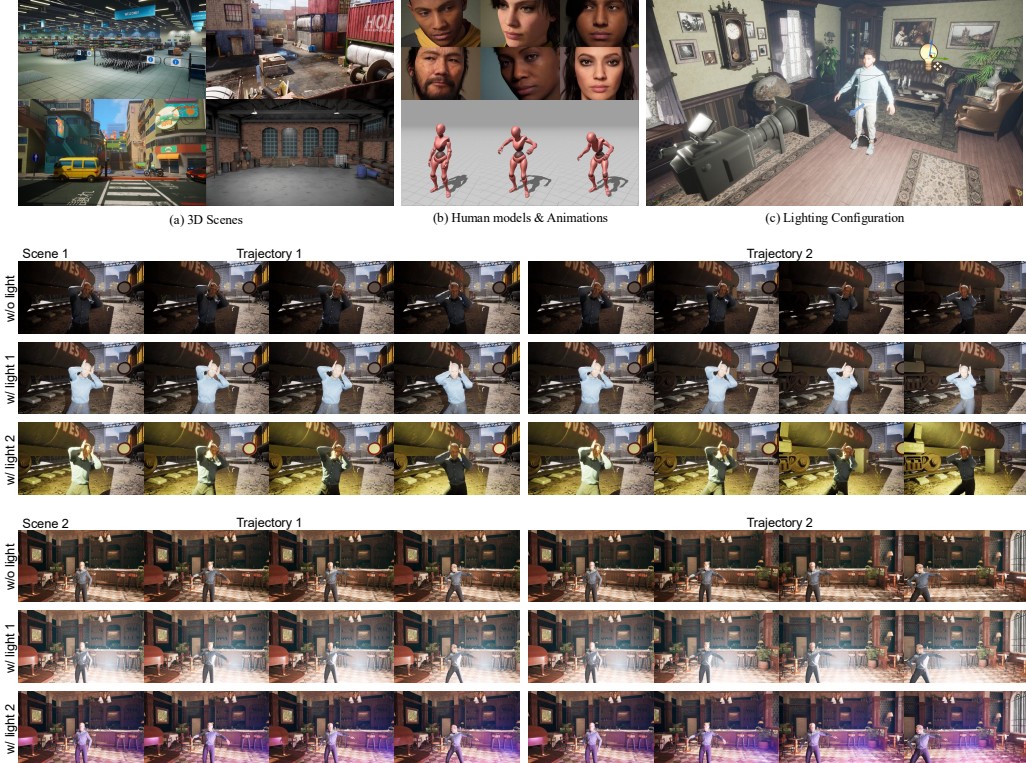

Figure 1: **Dataset Overview.** (a) and (b) show the assets used in our relighting datasets, including the 3D scenes, human models, and animations. (c) demonstrates an example lighting configuration. For each scene that has been set up, denoted as w/o light, we sample multiple camera trajectories and additional light sources to render video editing pairs with diverse motion and light conditions.

light source refers to a light source whose parameters can change over time. We develop a simple rule to generate three batches of random data to enhance data diversity. 1) Fixed light source with a fixed depth slightly behind the camera's initial position. 2) Fixed light source with fully random parameters. 3) Variable light source with fully random parameters, and one of these parameters can further change over time. e.g., 2D coordinates, depth, color, or intensity.

We obtained a total of 7,824 pairs of training data through Unreal Engine 5 rendering, including the original video, the target video, and the corresponding original parameters of the lighting conditions. Each video has a resolution of 384x672 and a total of 77 frames. The prompts are generated based on the original videos using a common video caption model to eliminate the influence of the text content on lighting control. During the training process, the T2V base model can only generate target videos based on the given lighting conditions.

## 4 METHOD

A point light source contains three attributes: position $\mathbf{p} \in \mathbb{R}^3$, color $\mathbf{c} \in \mathbb{R}^3$, and intensity $I \in \mathbb{R}$. Considering that the light source may change over time, we also need a representation for temporally-varying lights. An intuitive solution to represent light sources is text, but precise lighting editing via text is difficult, as pretrained text-to-video (T2V) models have never seen such captions. Motivated by the Multi-plane Image (MPI) representing a 3D scene via multiple images at different depths, we propose a novel Multi-plane Light Image to encode 3D light information in a scene, including positions, colors, and intensities of multiple light sources. Specifically, we use 4 light images in an MPLI and compress the 4 images into one video latent feature via Video VAE, which is injected into DIT through a Light Image Adapter (LIA). For an input video of $4N + 1$ frames, we use $N$ MPLIs to represent temporally-varying scene lighting, which naturally aligns with the pretrained DIT (Peebles & Xie, 2023). We first brief on the preliminary knowledge of the pretrained T2V model, and then elaborate on our proposed MPLI and LIA.

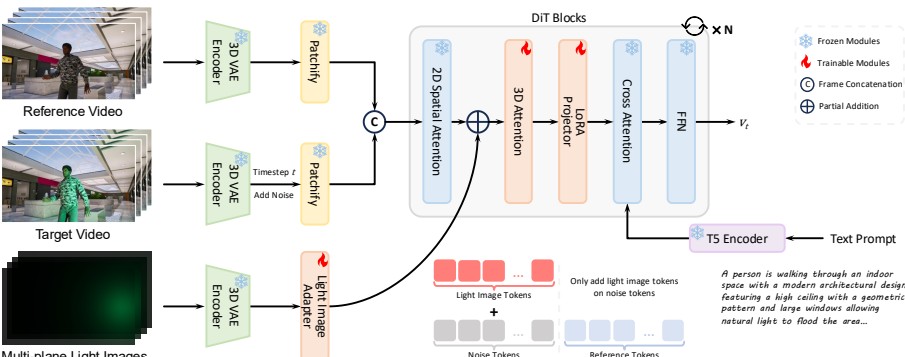

Figure 2: **An overview of our relighting training framework.** A Multi-plane Light Image (MPLI) contains 4 light images, and each MPLI is encoded as a latent light feature by the Video VAE. $N$ latent light features are passed to the DiT model via our proposed Light Image Adapter (LIA), which is initialized by the pretrained patchify module and shared across different DiT blocks. The original video and the noise are temporally concatenated. The parameters of the pretrained DiT model are frozen except the 3D attention layers. We also add a LoRA module after the 3D attention layer to learn the additional editing knowledge.

## 4.1 PRELIMINARY

We finetune our RelightMaster on a pretrained internal text-to-video (T2V) generation model, which adopts a latent video generation architecture. Given a video with $4N + 1$ frames, the T2V model pads 3 dummy images to the video and compresses 4 images as a latent video feature via a video variational encoder (VAE) (Kingma & Welling, 2013). Then the model is trained by the conditional flow matching loss (Lipman et al., 2022). We obtain the noisy video feature $x_t = (1-t)x_0 + t\epsilon$ at the timestep $t$ by interpolating the clean video latent feature $x_0$ and a noise sampled from the standard Gaussian distribution $\epsilon \in \mathcal{N}(0, 1)$ according to the timestep $t$, which corresponds to the ordinary differential equation (ODE): $dx_t = v_\Theta(x_t, t)dt$. The T2V model predicts the velocity $v_\Theta(x_t, t)$:

$$\mathcal{L}_{FM} = \mathbb{E}_{t, x_0, \epsilon} \big\| v_\Theta(x_t, t) - u_t(x_0 | \epsilon) \big\|_2^2. \tag{1}$$

In the inference stage, the T2V model uses the Euler scheduler to generate a video from noise:

$$x_t = x_{t-1} + v_\Theta(x_{t-1}, t) \cdot \Delta t. \tag{2}$$

$t$ iterates from 0 to 1.

## 4.2 MULTI-PLANE LIGHT IMAGE REPRESENTATION

Compared to environment maps that only characterize ambient light captured from real environments, our goal is to re-light the environment with new light sources in the 3D scene, which requires accurately injecting the 3D positions of the newly added light sources into the video generation network. Inspired by Multi-Plane Image (MPI), which uses multiple images at different depths to represent 3D scenes, we propose a Multi-Plane Light Image (MPLI) representation to encode the 3D positions of point light sources. Below, we first introduce the basic concept of a Light Image and then extend it to the multi-plane form.

**Light Image**. A Light Image is a normalized irradiance image rendered from light sources. Specifically, we place a plane orthogonal to the camera's orientation and pass through the camera's optical center, with a depth of $d$. Each pixel $(x, y)$ on this plane corresponds to the 3D position $\mathbf{q} = (x, y, d)$ in the camera coordinate system. To simplify modeling, we approximate the point light source as a single luminous particle with an illumination intensity of I, and its 3D position in the camera coordinate system is $\mathbf{p}_l = (x_l, y_l, z_l)$. In a homogeneous medium, the luminous intensity follows the inverse-square law: the intensity at a distance $r$ from the light source is inversely proportional to the square of $r$, i.e., $I_r \propto 1/r^2$. Thus, the irradiance at pixel on the Light Image is simplified as: $I_{x,y} = \frac{I}{\|\mathbf{q} - \mathbf{p}\|_2^2}$. To align with the video resolution, we crop an $H \times W$ region centered on the camera's optical center from the photosensitive plane. Suppose there are multiple light sources, we

sum up all the irradiance to obtain the entire information. Additionally, to adjust the numerical range of irradiance for better adaptation to the video generation network, we introduce two scalers $s_1$ and $s_2$ to modify the above equation:

$$I_{x,y} = \sum_i \frac{I_i \cdot \mathbf{c}_i}{||\mathbf{q} - \mathbf{p}_i||_2^2 / s_1 + s_2}. \tag{3}$$

$i$ indicates the $i$-th light source. The final RGB lighting information captured on the light image is obtained by multiplying the irradiance $I_i$ with the color of the point light source $\mathbf{c}_i$.

**Multi-Plane Light Image**. The single-plane light image can only encode the projection of light sources onto a fixed depth plane, failing to distinguish the 3D depth of different point light sources. To address this, we extend the single photosensitive plane to multiple parallel photosensitive planes as the Multi-Plane Light Image (MPLI), where each plane corresponds to a unique depth in the camera coordinate system. In our experiments, we set up four parallel photosensitive layers, each with a distinct depth. We use multiple MPLIs to further support light sources varying over time. Each MPLI corresponds to a video moment. Sequentially arranged, these MPLIs accurately capture dynamic changes (position, intensity) and meet video generation's lighting coherence needs.

### 4.3 LIGHT IMAGE ADAPTER

To enable effective injection of Multi-plane Light Image (MPLI), which efficiently encodes multi-source lighting in scenes, into the pre-trained text-to-video (T2V) pipeline, we further propose a Light Image Adapter (LIA). Our pre-trained T2V model processes videos with $4N + 1$ frames: after padding 3 dummy frames, a video Variational Autoencoder (VAE) compresses every 4 consecutive frames into a single video latent feature. To align with this architectural design, we set $K = 4$ for MPLI, such that a single MPLI can be compressed by the same pre-trained Video VAE into a latent light feature, matching the dimensionality and distribution of video latents. To support temporally-varying lighting, i.e., light sources varying across video frames, we associate one MPLI with each 4-frame interval in the input video. Thus, for a video undergoing relighting, we configure $N$ MPLIs in total. While this lighting representation operates at a 4-frame granularity rather than per-frame, we find it sufficient for most scenarios, since the Diffusion Transformer (DiT) model inherently smooths lighting effects for intermediate frames.

We propose LIA to inject sequential MPLIs into the network while preserving the pre-learned video prior. Specifically, the pre-trained T2V model first passes video latent features through a patchify module for further compression. To ensure compatibility with the learn video prior to the T2V model, our LIA reuses the structure of the patchify module and initializes its parameters with those of the pre-trained patchify module. After encoding the latent light feature, the LIA injects this signal into each DiT block. Critically, LIA parameters are shared across all blocks. We find this parameter-sharing mechanism, which plays as a form of self-regularization, crucial, as it mitigates overfitting, which is an issue that frequently arises when introducing new control modalities without such regularization. Besides LIA, we finetune the 3D attention in the pretrained T2V model to accommodate the increased token sequence length and add a low rank (LoRA) projector to absorb the additional lighting knowledge.

## 5 EXPERIMENTS

In this section, we first provide a series of experiments to show the controllability of our RelightMaster, and then we compare our RelightMaster with other state-of-the-art video relighting methods to show the superiority. Finally, we present ablation studies to show the effectiveness of our proposed Multi-plane Light Image and Light Image Adapter.

### 5.1 CONTROLLABLE VIDEO RELIGHTING

To comprehensively evaluate the effectiveness of our proposed RelightMaster in handling diverse lighting conditions for video relighting, we design a series of controlled experiments. Our Relight-Master generates relit videos according to the input lighting conditions, which include light source

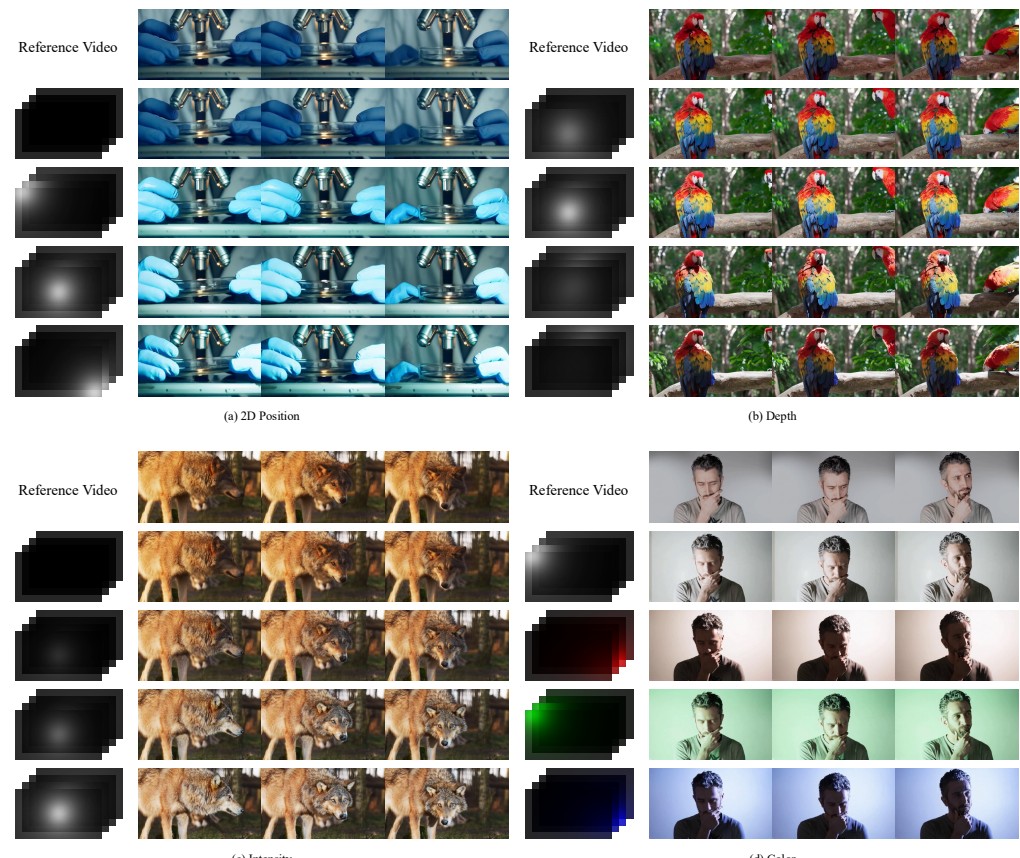

Figure 3: **Relighting with fixed light source.** (a) and (b) demonstrate the light source position, (c) reflects the light source intensity, and (d) indicates the light color.

positions in the 3D camera frustum, and the light color and intensity. We thus individually control the light conditions and relight the query videos in Fig. 3 to demonstrate precise controllability.

**Light source position**. We conduct two experiments that correspond to Fig. 3 (a) and (b), respectively. In experiment (a), we use a baseline video with no additional light sources to preserve the original appearance of the input video. We then compare this baseline against relit videos that use a single point light source with fixed depth. This point light source is placed at three distinct 2D positions: top-left, center, and bottom-right. The relit videos exhibit position-dependent specular reflections. Specifically, visible highlight regions appear on the rubber gloves of the dynamic object, and these highlights align with the 2D positions of the applied light sources. In experiment (b), we fix the 2D position of the point light source to the center and then gradually increase the depth of the light source. We generate four relit videos with increasing depth values, and each video shows distinct lighting effects. The shallow depth produces frontal low-intensity illumination. The moderate depth creates side illumination, and the large depth, with the light source behind the scarlet macaw, results in backlighting. These results confirm that the model accurately responds to adjustments in light source position, enabling fine-grained control over 3D lighting position.

**Light intensity**. In Fig. 3 (c), we fix the 3D position of a white point light source and gradually increase the light intensity starting from 0, equivalent to no additional light, to higher values, generating a sequence of relit videos with incremental intensity levels. The relit videos exhibit a clear correlation with the increasing light intensity. The wolf's head and body are gradually brightened by the white light as the intensity rises. Concurrently, the cast shadows also become progressively stronger with higher intensity. Such lighting effects reflect the model's accurate response to light intensity adjustments.

**Light color and position**. In this experiment, we fix the 3D position of the point light source to a side-lighting configuration and keep its intensity constant at a moderate level to avoid overexposure. We then test four distinct light colors: white, red, green, and blue, generating a separate relit video

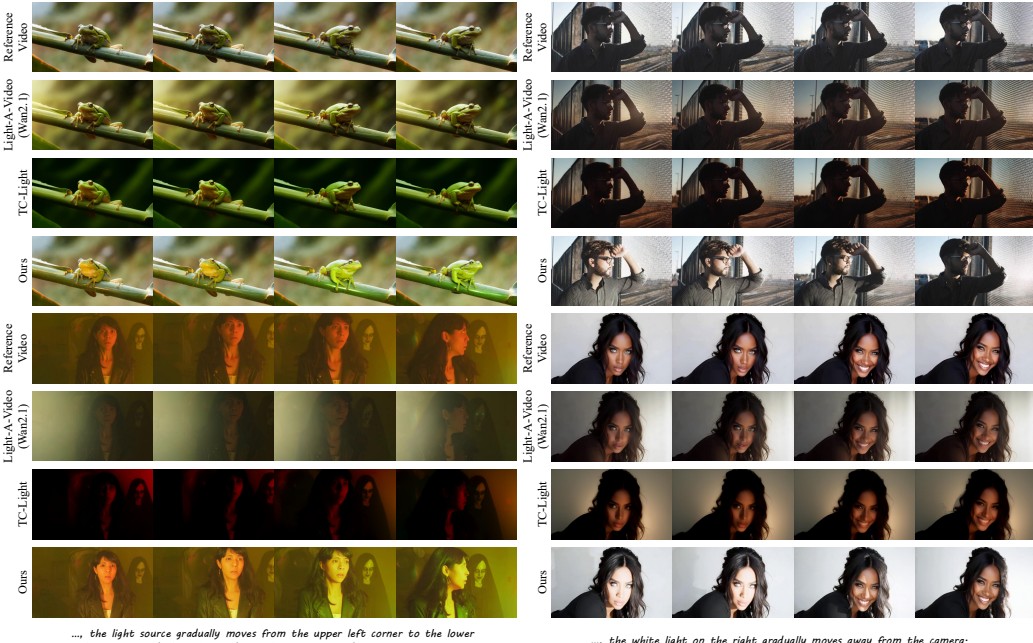

Figure 4: **Relighting with temporally-varying lights and multi-lights**. Our RelightMaster supports multiple and temporally-varying light source control. The corresponding Multi-plane Light Images (MPLI) at different moments are visualized for better understanding.

..., the light source gradually moves from the upper left corner to the lower right corner, and the color gradually changes from red to green.

..., the white light on the right gradually moves away from the camera.

Figure 5: **Comparison with other video relighting methods.** We translate our precise light control signals to text and feed them to Light-A-Video (Zhou et al., 2025) and TC-Light (Zhou et al., 2025). for each color. As shown in Fig. 3 (d), each relit video exhibits color-specific lighting effects that align with the applied light color. Specifically, on the male subject in the video, the face, hair, and clothing folds all show corresponding color-cast. Moreover, across all color settings, natural shadows, which are consistent with the side-lighting position, and diffuse reflections are clearly observed. These natural and color-accurate lighting effects reflect the precise color control capability.

**Temporally-varying lights and multi-lights**. In Fig. 4, over the duration of the video corresponding to the flower, we apply two temporal variations: 1) the light source moves continuously from the top-left to the bottom-right and 2) the light color transitions smoothly from red to green. As observed in the relit video, the flower's petals and stamens show color casts shifting from red to green, while the highlight and shadow positions on the flower surface follow the light's movement. For the video of the man, we deploy a blue light and a green light in the scene. We make the blue light intensity stronger and the green light weaker. The relit video accurately responds to the light variations. These results denote that our RelightMaster can precisely synchronize temporal adjustments of light position and color and support multiple lights.

## 5.2 COMPARISON WITH STATE-OF-THE-ART VIDEO RELIGHTING

We choose Light-A-Video (Zhou et al., 2025) and TC-Light (Liu et al., 2025b) for comparison. Light-A-Video and TC-Light extend the light control capability of IC-Light (Zhang et al., 2024)

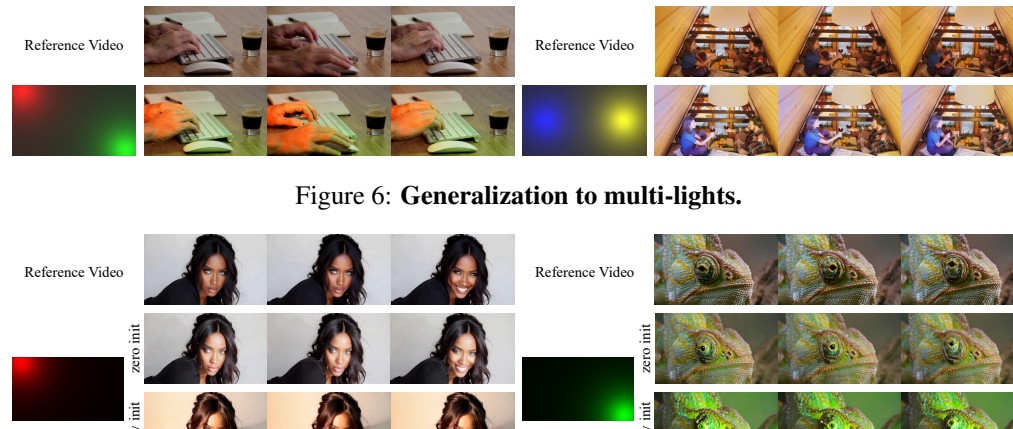

Figure 6: **Generalization to multi-lights.**

Figure 7: **Light Image Adapter Initialization.** "zero init" and "copy init" respectively denote that the LIA is initialized with a zero convolution or the parameters from the pretrained patchifier.

from images to videos. Similar to IC-Light, they rely on text prompts and an environment map to regulate relighting effects. However, the use of an environment map will replace the background of the original image or video, which is unacceptable for scenarios requiring background preservation. We thus transform the relighting conditions into text descriptions and append these descriptions to the original video caption and feed the merged prompt to Light-A-Video to obtain its relighting results. For our RelightMaster, we use the Multi-plane Light Images (MPLI) to indicate the relighting conditions. We conduct four experiments for comparison, as shown in Fig. 5. Two experiments focus on dynamic light position and color changes: the light source is moved from the top-left to the bottom-right of the frame, with its color gradually transitioning from red to green. The remaining two groups involve dynamic depth adjustment of a white light source, which is gradually moved forward along the camera lens. We compare our RelightMaster against Light-A-Video and TC-Light to reveal the clear performance gaps. Light-A-Video and TC-Light show no response to the relighting conditions. In contrast, our RelightMaster accurately responds to the instructions. In the position-color transition experiments, the light source moves smoothly from the top-left to the bottom-right, with the color gradually shifting from red to green. In the other experiments where the white light source moves forward, a dynamic and physically consistent lighting process is observed on the male and female subjects: initially, the white light brightens them by direct illumination. As the light continues to advance past the subjects' lateral position, the subjects begin to be partially occluded by their own contours, resulting in subtle shadow. Finally, the light moves further forward to the back of the object. The subjects exhibit a clear backlighting effect, indicating that our RelightMaster clearly outperforms the other methods.

## 5.3 Ablation Study

We provide two ablation studies with a single Light Image, i.e., $K = 1$, and on a 1/3 training dataset. As shown in Fig. 6, trained only on the single light source relighting data, our method can generalize to multi-source light source relighting, which reveals the extraordinary generalization performance of our Light Map representation. A common strategy used in image and video adapters is to initialize the parameters with a zero-convolution. However, the zero-initialization technique can not activate the relighting controllability (Fig. 7). In contrast, we initialize our Light Image Adapter with the parameters from the patchifier enabling video relighting, which reveals the significance that aligns the lighting control signals to the prior distribution learned by the DiT.

## 6 Conclusion

We proposed a novel framework RelightMaster for video relighting, which includes a dataset RelightVideo, a Multi-plane Light Image (MPLI) for accurate light source control, and a Light Image Adapter (LIA) for light feature injection. The experiments demonstrated that RelightMaster is able to individually control the light source position, color, and intensity for video relighting.

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

# A  APPENDIX

**LLM usage**. We used Gemini to help us polish our writing.

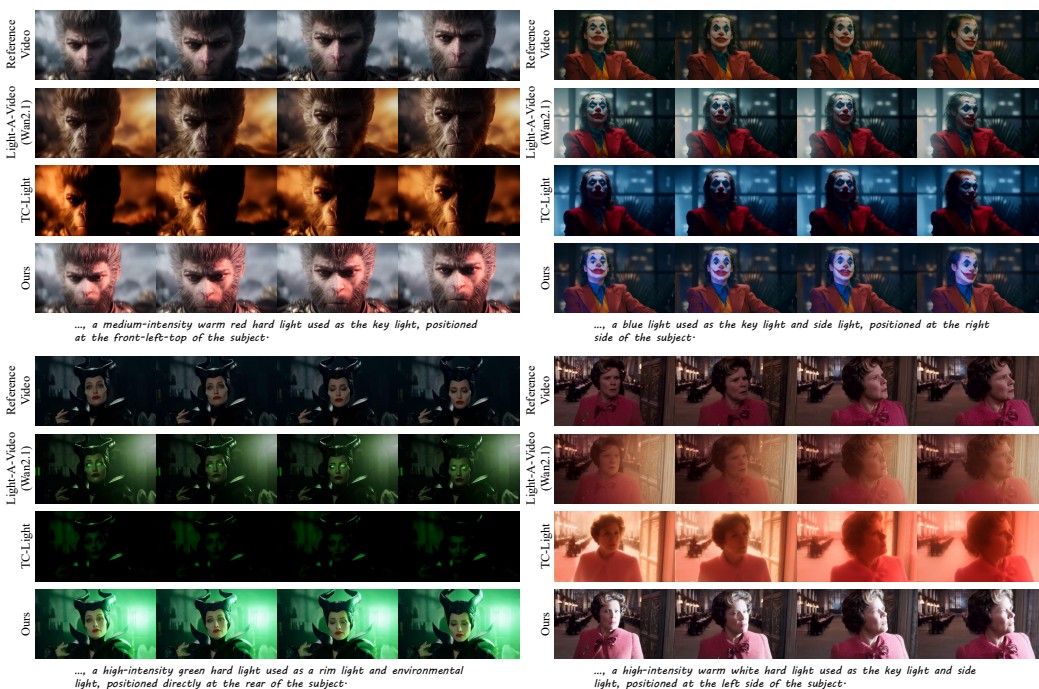

Figure 8: **Visualization for the quantitative comparison**

Table 1: **Quantitative Comparison on Synthetic Dataset**

| METHOD | PSNR ↑ | SSIM ↑ | LPIPS ↓ |
|---|---|---|---|
| Light-A-Video | 12.984 | 0.621 | 0.338 |
| TC-Light | 10.534 | 0.432 | 0.525 |
| Ours | **19.456** | **0.808** | **0.157** |

Table 2: **Quantitative comparison on Real Videos with VBench Metrics**

| METHOD | Subject Consistency ↑ | Background Consistency ↑ | Motion Smoothness ↑ | Dynamic Degree ↑ | Aesthetic Quality ↑ | Imaging Quality ↑ |
|---|---|---|---|---|---|---|
| Light-A-Video | **0.960** | **0.972** | 0.993 | **0.4** | **0.604** | 0.479 |
| TC-Light | 0.953 | 0.961 | **0.996** | **0.4** | 0.549 | 0.351 |
| Ours | 0.958 | **0.972** | 0.994 | **0.4** | 0.598 | **0.612** |

**Quantitative Comparison on Synthetic Dataset** To enable a direct comparison against Ground Truth (GT), we constructed a synthetic test set following a pipeline similar to our training data generation. We rendered 10 video pairs featuring challenge single-light sources, consisting of paired pre-edit (input) and post-edit (GT) sequences. We evaluated the performance of our method against Light-A-Video and TC-Light using standard image quality metrics: SSIM, PSNR, and LPIPS, in Tab. 1. Our method demonstrates superior performance across these metrics, indicating the highest fidelity to the target lighting effects compared to the other methods.

Table 3: **Quantitative Comparison on the User Study Results**

| METHOD | Video Quality ↑ | Relight Controllability ↑ | Content retention ↑ |
|---|---|---|---|
| Light-A-Video | 0 | 2 | 2 |
| TC-Light | 0 | 0 | 0 |
| Ours | **130** | **128** | **128** |

Table 4: **Ablation Study on K with Video Reconstruction Metrics.**

| K | PSNR ↑ | SSIM ↑ | LPIPS ↓ |
|---|---|---|---|
| 2 | 23.546 | 0.866 | 0.113 |
| 4 | 23.456 | **0.869** | **0.105** |
| 8 | **23.581** | 0.866 | 0.107 |

**Quantitative Comparison on Real Videos.** To evaluate performance in diverse real-world scenarios, we curated a test set comprising 10 classic clips selected from various movies. We visualized 4 of them in Fig. 8. Since GT is unavailable, we compare the performance by video generation metrics and human evaluation. We utilized VBench to assess the general video generation quality as shown in Tab. 2. The "imaging quality" metrics indicate that the quality of videos generated by our RelightMaster significantly surpasses the other two methods. Regarding the aesthetic quality metric, RelightMaster performs comparably to Light-A-Video. However, in terms of actual human perception, our method demonstrates significantly better visual quality. We attribute this discrepancy to the inherent bias in the aesthetic operator. We also conducted a user study to evaluate editing accuracy (i.e., how well the lighting change matches the prompt/control) with these 10 videos. Participants were asked to select the best results from the generated videos in terms of three questions:

- Which video presents the best quality?
- Which video presents the best relighting controllability?
- Which video best preserves the content of the input video?

There are 13 people participating in the user study. We collect the scores for each of the three methods across the three questions as shown in Tab. 3. The results present that our RelightMaster dominantly outperforms the other three methods. The quality of videos generated by Light-A-Video and TC-Light is not visually pleasing and the controllability is not good enough, as shown in Fig. 8. In contrast, the videos generated by our RelightMaster present good aesthetics and fine-grained controllability, which explains why most users rank videos generated by our RelightMaster 1st in the user study.

Table 5: Ablation Study on K with VBench Metrics

| K | Subject Consistency ↑ | Background Consistency ↑ | Motion Smoothness ↑ | Dynamic Degree ↑ | Aesthetic Quality ↑ | Imaging Quality ↑ |
|---|---|---|---|---|---|---|
| 2 | **0.930** | 0.939 | **0.993** | 0.563 | **0.538** | 0.602 |
| 4 | **0.930** | **0.941** | **0.993** | **0.594** | 0.546 | 0.609 |
| 8 | 0.929 | 0.939 | **0.993** | **0.594** | 0.534 | **0.611** |

**Ablation Study on K.** We provide an ablation study to show the reason why we select $K = 4$. We render 32 video pairs that contain temporally varying light sources to evaluate the performance. We compare the models with different numbers of multi-plane light maps (K) with the video reconstruction metrics in Tab. 4 and video quality metrics (VBench) in Tab. 5. We observe significant performance improvement, including LPIPS in the Tab. 4 and dynamic degree and aesthetic quality in Tab. 5, from $K = 2$ to $K = 4$, but the performance of $K = 4$ and $K = 8$ is comparable. We thus select $K = 4$ as our final setting.

