# OpenReview forum: "RelightMaster: Precise Video Relighting with Multi-plane Light Images"
_ICLR.cc/2026/Conference — Submitted to ICLR 2026_

### Official Review · Reviewer_TeRx · 2025-10-27

**Soundness:** 3
**Presentation:** 3
**Contribution:** 3
**Rating:** 6
**Confidence:** 4

**Summary:**

The paper presented a RelightMaster method which introduced Multi-Plane Light Image (MPLI) to allow precise description of lighting conditions at different depths for video relighting. RelightMaster uses 4 planes of light images at 4 depths to model the irradiance in a 3D scene by a light source, which allows finer and more precise lighting condition than using text descriptions. These conditions are injected to a private T2V generation model by a Light Image Adapter (LIA). To train this model, a new synthetic RelightVideo dataset collects 7,824 pairs of videos in the same scene with different lighting conditions generated by Unreal Engine 5. The proposed RelightMaster compares favorably with Light-A-Video and TC-Light.

**Strengths:**

The major technical contribution of this paper is the new MPLI representation which does allow more precise and finer description of the irradiance in a 3D scene. Thus, it demonstrates clear advantages on controlling the lighting in terms of light source position, intensity, and changing light sources and multi-lights.

The LIA is a reasonable scheme to integrate MPLI into the T2V model. The way to collect the RelightVideo dataset may be inspiring to other relighting works.

**Weaknesses:**

The major concern is on the experiments which are hard for other researchers to reproduce and compare with, since RelightMaster is built on a private T2V generation model. MPLI introduces more detailed lighting conditions than text description, so it is expected to outperform competing methods after these light control signals are translated to vague text prompts and feed to Light-A-Video and TC-Light.

The ablation study is quite simple, just showing the performance of K=1 and with 1/3 training dataset. It is not clear what the contribution of the private T2V model and the RelightVideo dataset are. In the ablation, is it possible to replace the private T2V model with one that is publicly available to other researchers? Is it possible to finetune other relighting method with the RelightVideo dataset?

**Questions:**

If RelightVid and Lumen are closely related work to compare with?

Did I miss the session of Reproducibility Statement?

---

> ### Author Response · Authors · 2025-12-03
> **Response to Reviewer TeRx**
>
> > **W1&W2a&Q2. Reproducibility and open-source plan.**
>
> We are currently adapting our method to WAN. The **dataset and model weights will be open-sourced** upon publication to ensure full reproducibility.
>
> > **W2b. Is it possible to finetune other relighting method with the RelightVideo dataset?**
>
> Yes, but they are still not able to fulfill fine-grained relighting control. RelightVideo provides the original video, the relight video, and the light source configurations. Finetuning the other relighting methods with the original video and the relight video is possible, but no other relighting methods are able to accept the 3D light source signals. This is also the motivation of our MPLI and LIA.
>
> > **Q1. If RelightVid and Lumen are closely related work to compare with?**
>
> They are closely related works but cannot be compared with. We failed to reproduce the result of RelightVid, which is not a published work. Its framework is similar to the Light-A-Video and TC-Light, which are all extensions of IC-Light, and we have included them for comparison. We cannot compare our method with Lumen because it will replace the background. We include these two works in the related work section in the paper and update the discussion.

---

### Official Review · Reviewer_53nu · 2025-10-30

**Soundness:** 3
**Presentation:** 3
**Contribution:** 3
**Rating:** 6
**Confidence:** 4

**Summary:**

The paper proposes RelightMaster, a diffusion-based framework for controllable video relighting. The goal is to achieve fine-grained lighting control using visual prompts instead of text.

The work identifies two key limitations of existing T2V models:

1. text prompts cannot effectively describe lighting details.

2. real controllable lighting data is scarce.

The authors introduce three main components:

1. RelightVideo Dataset, a synthetic relighting dataset rendered with Unreal Engine ensuring consistent scene content under varying light conditions

2. Multi-Plane Light Image (MPLI), a novel visual representation of lighting using K depth-aligned planes encoding 3D light source position, color, and intensity

3. Light Image Adapter (LIA), injects compressed MPLI features into pre-trained Video Diffusion Transformers (DiT) using the existing VAE encoder weights for better alignment with the video prior. Experiments show controllable relighting effects across light position, intensity, and color, and demonstrate multi-source and temporally-varying relighting control.

**Strengths:**

1. Addresses a clear gap in fine-grained video relighting

2. Physically interpretable representation of lighting through MPLI

3. The use of synthetic paired data is well justified.

4. The framework demonstrates convincing qualitative results for position, color, intensity and depth control.

**Weaknesses:**

1. Directional modeling: The paper assumes isotropic point lights defined by position, color, and intensity only. Modeling directional or area lights would improve realism and generality. Future work should explore non-isotropic light sources.

2. Quantitative evaluation: No numerical comparison is provided. It would be important to include metrics such as (a) relighted image quality (FID or CLIP-based), (b) temporal consistency, and (c) user preference. Qualitative evaluation alone is insufficient for assessing generalization. Earlier works like Light-A-Video have provided quantitative metrics.

3. MPLI design: MPLI uses 4 planes aligned with the camera to match the Video VAE compression scheme. However, it would be worth exploring having 4 light planes per frame (rather than per latent) and fusing them, as this could capture finer lighting variation over time. An ablation on the number of planes (K) is missing.

4. Evaluation scope: Comparisons to Light-A-Video and TC-Light rely on textual relighting descriptions, which may not be fully fair given those methods were not designed for fine spatial control.

**Questions:**

Overall: This is a strong paper with clear novelty, solid technical grounding, and compelling qualitative results. Addressing quantitative evaluation and exploring directional lighting or richer MPLI structures would further strengthen it.


1. How sensitive are results to the number of light planes (K)?

2. Could directionality or angular falloff be included in MPLI without destabilizing training?

3. Can quantitative evaluation be added (even on synthetic data) to measure temporal consistency or realism?

---

> ### Author Response · Authors · 2025-12-03
> **Response to Reviewer 53nu**
>
> > **W1. Future work should explore non-isotropic light sources.**
>
> Yes. We would explore more complex light conditions, such as non-isotropic light sources in the future. A straightforward solution is to introduce light directions as an embedding in MPLI. Direction embedding has been proven effective in CameraCtrl [1].
>
> > **W2. Quantitative evaluation.**
>
> Please refer to the general response.
>
> > **W3. MPLI design: it would be worth having 4 light planes per frame (rather than per latent) and fusing them.**
>
> Thanks for your suggestion! Compressing 4 frames into 1 frame means that the granularity of lighting representation is 6fps (the original video is generally 24fps). We have observed that the lighting in most videos is relatively smooth, and 6fps of lighting information is sufficient to express the light information. We will try to adopt the fusing strategy to support more fine-grained relight in the future.
>
> > **W4. Evaluation scope：Comparisons to Light-A-Video and TC-Light rely on textual relighting descriptions, which may not be fully fair.**
>
> Our goal is to address **fine-grained**, **3D-controllable video relighting**. To the best of our knowledge, our work is the first solution for this task.  Light-A-Video and TC-Light can only be controlled via text prompts, which are not able to address this task. It further underscores the **significance of our RelightMaster framework.**
>
> > **Q1. How sensitive are results to the number of light planes (K)?**
>
> Please refer to the general response.
>
> > **Q2. Could directionality or angular falloff be included in MPLI without destabilizing training?**
>
> Direction embedding has been proven effective in CameraCtrl [1]. We believe that including it in MPLI will not destabilize training. We will explore the directionality or angular falloff in the future.
>
> > **Q3. Can quantitative evaluation be added (even on synthetic data) to measure temporal consistency or realism?**
>
> Please refer to the general response.
>
> > **Reference**
>
> [1] He, Hao, et al. "Cameractrl ii: Dynamic scene exploration via camera-controlled video diffusion models." arXiv preprint arXiv:2503.10592 (2025).

---

### Official Review · Reviewer_zrp3 · 2025-10-31

**Soundness:** 1
**Presentation:** 3
**Contribution:** 2
**Rating:** 4
**Confidence:** 4

**Summary:**

This paper introduces a novel framework for precise video relighting, addressing key challenges in lighting control within video generation. The authors propose a Multi-plane Light Image (MPLI) representation to encode 3D light source properties and a Light Image Adapter (LIA) to inject this information into pre-trained video diffusion models. A synthetic dataset, RelightVideo, is constructed using Unreal Engine to enable training with consistent dynamic content under varying lighting conditions. Experiments demonstrate fine-grained control over light position, intensity, color, and temporal variation, outperforming existing methods in both accuracy and generalization.

**Strengths:**

1.	The paper proposes MPLI representation effectively encodes 3D light source properties and naturally aligns with video modality, enabling precise and dynamic multi-light control.
2.	The Light Image Adapter offers a lightweight and efficient integration strategy that preserves pre-trained model knowledge and avoids catastrophic forgetting.
3.	The RelightVideo dataset provides the first large-scale video relighting dataset with consistent dynamic content under varied lighting conditions.

**Weaknesses:**

1.	The experimental section lacks quantitative results and metrics, relying solely on qualitative visual examples, which makes a rigorous performance comparison impossible.
2.	The comparison with state-of-the-art methods is limited to only two other approaches (Light-A-Video and TC-Light) and relies solely on qualitative visual examples, with no quantitative metrics or user studies provided to substantiate the claimed superiority.
3.	The method's performance and generalization on real-world videos, which have more complex lighting and textures than the synthetic training data, are not thoroughly evaluated or discussed.

**Questions:**

1.	In the ablation study on LIA initialization, what is the specific performance drop when using "zero init" compared to "copy init"?
2.	The paper uses a fixed number of four planes (K=4) for the MPLI representation,it was this value determined empirically?
3.	The method is trained exclusively on synthetic data from Unreal Engine,how does it generalize to real-world videos with complex natural lighting and textures?
4.	Is there a plan to open source the dataset? This is very important for reproducing the results of the paper.

---

> ### Author Response · Authors · 2025-12-03
> **Response to Reviewer zrp3**
>
> > **W1&W2&W3. The experimental section lacks quantitative results and metrics.**
>
> Please refer to the general response.
>
> > **Q1. In the ablation study on LIA initialization, what is the specific performance drop when using "zero init" compared to "copy init"?**
>
> As shown in the Fig. 7 in the paper, the "zero init" module cannot respond to the light source.
>
> Simple:
>
> |  Method | PSNR $\uparrow$ | SSIM $\uparrow$ | LPIPS $\downarrow$ |
> | --- | --- | --- | --- |
> | zero init | 24.452 | 0.882 | 0.090 |
> | copy init | **26.521** | **0.895** | **0.073** |
>
> Challenging:
>
> |  Method | PSNR $\uparrow$ | SSIM $\uparrow$ | LPIPS $\downarrow$ |
> | --- | --- | --- | --- |
> | zero init | 23.601 | 0.872 | 0.100 |
> | copy init | **25.964** | **0.879** | **0.092** |
>
> We further conduct a quantitative ablation study as shown above.  Following a pipeline similar to our training data generation, we build a test set that provides the input video with the corresponding groundtruth relight video. The test set consists of two subsets, i.e., a simple set where the camera is static, and a challenging set where the camera moves in the environment. We compute the PSNR, SSIM, and LPIPS metrics for the generated video and the ground truth for evaluation. As shown above, **"copy-init" significantly outperforms "zero-init" all-sided**.
>
> > **Q2. The paper uses a fixed number of four planes (K=4) for the MPLI representation, was this value determined empirically?**
>
> Please refer to the general response.
>
> > **Q3. The method is trained exclusively on synthetic data from Unreal Engine, how does it generalize to real-world videos with complex natural lighting and textures?**
>
> 1. The **MPLI and LIA design** ensures that the Light tokens remain closely aligned with the original video tokens' feature distribution. This forces the network to focus primarily on learning the **lighting differential** rather than relearning video generation from scratch. This philosophy is directly supported by our "zero init v.s. copy init" ablation, which shows the strong benefit of reusing existing video features.
> 2. Furthermore, we found that limiting the trainable parameters to the **3D self-attention layers** is a crucial strategy. Training the FFN layers significantly increases the risk of **style shift** and reduces generalization, which also reflects the importance of aligning to the distribution of the pretrained model.
>
> > **Q4. Is there a plan to open source the dataset? This is very important for reproducing the results of the paper.**
>
> Yes. The **dataset and model weights will be open-sourced** upon publication to ensure full reproducibility.

---

### Official Review · Reviewer_adZs · 2025-10-31

**Soundness:** 3
**Presentation:** 3
**Contribution:** 3
**Rating:** 4
**Confidence:** 2

**Summary:**

This paper proposes RelightMaster, a video relighting framework with a novel lighting representation Multi-plane Light Image (MPLI). To address the data scarcity of relighting videos, the authors also built a dataset called RelightVideo that features synthetic rendered videos using the Unreal Engine. MPLI can be injected into pre-trained video diffusion transformers to generate relighted videos with target lighting conditions. Qualitative results are reported with comparison to Light-A-Video and TC-Light.

Overall, I like the MPLI idea to represent lighting conditions, and the designed relighting framework is reasonable. On the other hand, the experiment section is weak, and the choice of using point light does not really show the potential of MPLI in the relighting task.

**Strengths:**

- The most interesting aspect of this paper is the proposed Multi-plane Light Image (MPLI) for lighting representation. It is extended from the idea of the Multi-Plane Image for 3D representation and adapted to lighting. Such a representation could be useful for a broader range of applications that utilize lighting.

- The RelightVideo dataset could be a good contribution, if it will be released to the public.

**Weaknesses:**

- While the idea of the Multi-plane Light Image is interesting, the current usage of it to represent lighting is limited. Take a look at Figures 3 and 4, the majority of the MPLI are empty (i.e., black). This seems to be a not-so-efficient way of representing *point lights*. MPLI should be able to capture much complex spatially and temporally varying lighting. I feel like this is a missed opportunity.


- There is no quantitative evaluation in the experiment section. I understand that evaluating relighting results in general is challenging, especially for videos. But there are still common metrics that other relighting methods report, such as PSNR. In the ideal case, the authors should have both a metrics-based evaluation and a user study. The current experiment is quite weak.

Minor thing
1. Caption for Figure 2: typo of 'dataset', it should be 'An overview of our relighting framework'. Also, the caption should mention that this overview is for training.

**Questions:**

1. Can the authors show some results with more complex lighting than point lights?

2. Can the authors discuss the efficiency of the MPLI to represent point lights? It seems most of the images are blank (for example, Figure 2).

3. Line 237 mentions that the authors use an 'internal text-to-video' model. Why not use an open source model such as WAN?

4. How does the method compare to simple frame-by-frame image-based relighting methods?

5. Can the authors show some quantitative results?

---

> ### Author Response · Authors · 2025-12-03
> **Response to Reviewer adZs**
>
> > **W1a: Majority of the MPLI are empty. MPLI is a not-so-efficient way of representing *point lights.***
>
> The MPLIs are not empty. The irradiance derived from the light sources diffuses to MPLIs, but the visualization is not prominent. The primary goal of MPLI is to explore an **effective and fine-grained representation** for video relighting, rather than achieving peak efficiency. Given the current lack of a solution for **fine-grained, temporally consistent** video relighting, we prioritize **effectiveness** in capturing complex spatio-temporal lighting over immediate efficiency.
>
> We design an ablation study to show the effectiveness of our MPLI.  We use the parameters of the point light sources, i.e., **3D position, color, and intensity**, to represent the light source and encode them following ReCamMaster [1] to learn the relighting effects. We build a test set that provides the input video with the corresponding groundtruth relight video following the training data generation and evaluate the methods with video reconstruction metrics.
>
> Simple:
>
> |  Method | PSNR $\uparrow$ | SSIM $\uparrow$ | LPIPS $\downarrow$ |
> | --- | --- | --- | --- |
> | Raw Params. | 21.443 | 0.782 | 0.151 |
> | MPLI | **26.521** | **0.895** | **0.073** |
>
> Challenging:
>
> |  Method | PSNR $\uparrow$ | SSIM $\uparrow$ | LPIPS $\downarrow$ |
> | --- | --- | --- | --- |
> | Raw Params. | 20.632 | 0.746 | 0.163 |
> | MPLI | **25.964** | **0.879** | **0.092** |
>
> This parametrization is the most efficient light representation. However, as shown above, the relighting effects is far behind MPLI.
>
> > **W1b: MPLI should be able to capture much complex spatially and temporally varying lighting.**
>
> Yes. Actually, MPLI has been able to capture complex spatially and temporally varying lighting. As demonstrated in **Figure 4 and the supplementary video**, we show examples of single or multiple light sources varying over time in their **3D position, color, and intensity**.
>
> > **W2: Quantitative evaluation**
>
> Please refer to the general response.
>
> > **W3: Caption for Figure 2**
>
> Thanks for the suggestion. We updated the paper to fix the typo.
>
> > **Q1: Can the authors show some results with more complex lighting than point lights?**
>
> As our RelightMaster is not trained on more complex lighting conditions, such as line light sources, the relighting effects of our RelightMaster are not good enough for them. We will explore video relighting with more complex lighting conditions in our future work.
>
> > **Q2: Discuss the efficiency of the MPLI to represent point lights? Most of the images are blank (for example, Figure 2).**
>
> Please refer to W1a and W1b.
>
> > **Q3:  Why not use an open source model such as WAN?**
>
> We used an internal text-to-video model initially due to business issues.
>
> We are currently adapting our method to WAN. The **dataset and model weights will be open-sourced** upon publication to ensure full reproducibility.
>
> > **Q4: How does the method compare to simple frame-by-frame image-based relighting methods?**
>
> Our baselines, **TC-Light** and **Light-A-Video**, are already video-enhanced versions of the SOTA image relighting method **IC-Light**. Simple per-frame application of IC-Light leads to significant **temporal flickering** and **illumination inconsistencies** across frames. This issue is already documented and shown in the original papers for TC-Light (Figure 3) and Light-A-Video (Figure 2).
>
> > **Q5. Can the authors show some quantitative results?**
>
> Please refer to the general response.
>
> > **Reference**
>
> [1] Bai, Jianhong, et al. "ReCamMaster: Camera-Controlled Generative Rendering from A Single Video" arXiv preprint arXiv:2503.11647 (2025)

---

### Author Response · Authors · 2025-12-03
**General Response**

We appreciate the reviewers' suggestion regarding the need for quantitative evaluations. As collecting paired ground truth (GT) data (i.e., the same video captured under different lighting conditions) in open-world settings is not feasible, we provide two quantitative comparisons with baselines, i.e., Light-A-Video and TC-Light, covering synthetic and real-world scenarios. We also provide an ablation study on the number of MPLI. **All the results are added to the paper's appendix**.

> **1. Quantitative Comparison on the Synthetic Test Set**

To enable a direct comparison against Ground Truth (GT), we constructed a synthetic test set following a pipeline similar to our training data generation. We rendered 10 video pairs featuring challenge single-light sources, consisting of paired pre-edit (input) and post-edit (GT) sequences. We evaluated the performance of our method against Light-A-Video and TC-Light using standard image quality metrics: **SSIM**, **PSNR**, and **LPIPS**.

Table 1.

| Method | PSNR $\uparrow$ | SSIM $\uparrow$ | LPIPS $\downarrow$|
| --- | --- | --- | --- |
| Light-A-Video (Wan2.1) | 12.984 | 0.621 | 0.338 |
| TC-Light | 10.534 | 0.432 | 0.525 |
| RelightMaster(Ours) | **19.456** | **0.808** | **0.157** |

As shown in the above Table, our method demonstrates superior performance across these metrics,  indicating the highest fidelity to the target lighting effects compared to the other methods.

> **2. Quantitative Comparison on Real Videos**

To evaluate performance in diverse real-world scenarios, we curated a test set comprising 10 classic clips selected from various movies. Since GT is unavailable, we compare the performance by video generation metrics and human evaluation:

Table 2.

| Method | Subject Consistency $\uparrow$ | Background Consistency $\uparrow$ | Motion Smoothness $\uparrow$ | Dynamic Degree $\uparrow$ | Aesthetic Quality $\uparrow$ | Imaging Quality $\uparrow$ |
| --- | --- | --- | --- | --- | --- | --- |
| Light-A-Video | **0.960** | **0.972** | 0.993 | **0.4** | **0.604** | 0.479 |
| TC-Light | 0.953 | 0.961 | **0.996** | **0.4** | 0.549 | 0.351 |
| Ours | 0.958 | **0.972** | 0.994 | **0.4** | 0.598 | **0.612** |

We utilized **VBench** to assess the general video generation quality as shown above. The ``imaging quality'' metrics indicate that the quality of videos generated by our RelightMaster significantly surpasses the other two methods. Regarding the aesthetic quality metric, RelightMaster performs comparably to Light-a-Video. However, in terms of actual human perception, our method demonstrates significantly better visual quality. We attribute this discrepancy to the inherent bias in the aesthetic operator.

We also conducted a user study to evaluate **editing accuracy** (i.e., how well the lighting change matches the prompt/control) with these 10 videos. Participants were asked to select the best results from the generated videos in terms of three questions:

1. Which video presents the best quality?
2. Which video presents the best relighting controllability?
3. Which video best preserves the content of the input video?

Table 3.

| Method | Video Quality $\uparrow$ | Relight Controllability $\uparrow$ | Content retention  $\uparrow$ |
| --- | --- | --- | --- |
| Light-A-Video | 0 | 2 | 2 |
| TC-Light | 0 | 0 | 0 |
| Ours | **130** | **128** | **128** |

There are 13 people participating in the user study. We collect the scores for each of the three methods across the three questions as shown above. The results present that our RelightMaster dominantly outperforms the other three methods.

We also update the paper appendix to visualize the results in the above quantitative comparison.

> **3. Ablation study on K**

We provide an ablation study to show the reason why we select $K=4$. We render 32  video pairs that contain temporally varying light sources to evaluate the performance. We compare the models with different numbers of multi-plane light maps (K) with the video reconstruction metrics:

Table 4.

| K | PSNR $\uparrow$ | SSIM $\uparrow$ | LPIPS $\downarrow$ |
| --- | --- | --- | --- |
| 2 | 23.546 | 0.866 | 0.113 |
| 4 | 23.456 | **0.869** | **0.105** |
| 8 | **23.581** | 0.866 | 0.107 |

and VBench metrics:

Table 5.

| K | Subject Consistency $\uparrow$ | Background Consistency $\uparrow$ | Motion Smoothness $\uparrow$ | Dynamic Degree $\uparrow$ | Aesthetic Quality $\uparrow$ | Imaging Quality $\uparrow$ |
| --- | --- | --- | --- | --- | --- | --- |
| 2 | **0.930** | 0.939 | **0.993** | 0.563 | 0.538 | 0.602 |
| 4 | **0.930** | **0.941** | **0.993** | **0.594** | **0.546** | 0.609 |
| 8 | 0.929 | 0.939 | **0.993** | **0.594** | 0.534 | **0.611** |

As shown above, we observe significant performance improvement, including lpips in Table 4 and dynamic degree and aesthetic quality in Table 5,  from $K=2$ to $K=4$, but the performance of $K=4$ and $K=8$ is comparable. We thus select $K=4$ as our final setting.

---

### Meta-Review · Area_Chair_Z7U7 · 2026-01-03

**Summary:**

The paper received mixed initial reviews, with scores of 6, 6, 4, and 4. Reviewers generally acknowledged the technical effort and found the idea of introducing an explicit, point-light–based lighting representation (MPLI) for video relighting to be interesting. The paper demonstrates convincing control over simple point-light configurations, and the additional quantitative evaluations and user study provided in the rebuttal improved the empirical support. At the same time, multiple reviewers raised substantive concerns regarding the limited scope of the lighting model and the fairness and relevance of the chosen baselines.

In the rebuttal, the authors addressed several technical and presentation-related issues by adding quantitative metrics, user studies, ablations on MPLI design choices, and commitments to open-sourcing data and adapting the method to an open-source backbone. These responses resolved some concerns about evaluation completeness and reproducibility. However, several core conceptual concerns remain only partially addressed. In particular, the method is limited to point-light illumination, while the primary comparisons are against text-driven relighting methods that are not designed to model explicit physical lighting parameters, but instead handle a much broader range of semantically complex lighting descriptions (e.g., sunsets, skylight, or sunlight through windows). This creates an inherent mismatch in the comparison setting, which weakens the conclusions drawn from the experimental results. The AC therefore anticipates that final reviewer opinions will likely remain mixed, with several reviewers maintaining reservations around borderline scores (see detailed discussion in Reviewer Concerns and Reviewer Scores).

The AC appreciates the proposed lighting representation and finds it interesting; however, they share the concerns highlighted by the reviewers regarding lighting expressiveness and comparison fairness. Moreover, despite aiming to introduce a novel explicitly parameterized lighting representation, the paper does not sufficiently position itself within the broader relighting literature that already leverages explicit physical lighting representations, such as spherical harmonics or environment maps. Many related works—such as Sun et al., “Single Image Portrait Relighting”, Pandey et al., “Total Relighting”, Jin et al., “Neural Gaffer”, and Cai et al., “Real-time 3D-aware Portrait Video Relighting”—are absent from the discussion, despite being highly relevant in spirit. While many of these methods focus on single images, some (e.g., Cai et al.) explicitly address video relighting and offer valuable points of comparison in terms of lighting representation and physical interpretability.

From a modeling standpoint, the proposed MPLI representation also appears fundamentally limited in its ability to represent directional or environment-map lighting, since the irradiance of distant illumination is governed by surface normals and angular variation rather than spatial variation. Under such conditions, MPLI would effectively collapse to a constant color and fail to capture directional lighting effects. While the method may be particularly well suited for spatially varying near-field lighting, this potential advantage is not convincingly demonstrated in the current experiments. More targeted evaluations—such as moving a point light along the viewing direction to emphasize near-field effects, or relighting with spatially complex emitters like LED panels, accompanied by clearer visualization of light positions—would more directly highlight the unique strengths of the representation and better justify its novelty. In contrast, most results presented in the paper involve static point lights or only slight light motion without explicit light parameter illustration; such effects could plausibly be approximated by static or rotating directional lights or environment maps. As a result, the uniqueness of the proposed representation is not fully established.

Overall, without a more comprehensive analysis of the proposed lighting representation, clearer positioning relative to existing explicitly parameterized relighting methods, and stronger evidence highlighting scenarios where the approach offers unique advantages, the AC finds it difficult to assess the true novelty and significance of the contribution. For these reasons, the AC recommends rejection, while encouraging the authors to more clearly analyze, position, and demonstrate the strengths of their explicitly controlled lighting representation in future work.

**Reviewer Concerns:**

### Reviewer adZs (Score: 4)

- The reviewer raised concerns about the lack of quantitative evaluation and user studies in the original submission, noting that the evidence relied heavily on qualitative results. They also questioned the limited lighting complexity (restricted to point lights), the potential inefficiency of the MPLI representation (with many empty regions for point lights), and the reliance on an internal T2V backbone.
- In the rebuttal, the authors added quantitative metrics on both synthetic and real videos (PSNR, SSIM, LPIPS, VBench), conducted a user study, and provided ablations comparing raw light parameters with MPLI. They clarified design choices related to sparsity and efficiency, noted support for more complex multi-light scenarios, and committed to open-sourcing the dataset and adapting the method to an open-source backbone.

---

### Reviewer zrp3 (Score: 4)

- The reviewer expressed concerns about the absence of quantitative evaluation and user studies, limited baseline comparisons, and the generalization of results to real-world videos given training on synthetic data. They also requested clarification on adapter initialization, the choice of the number of MPLI planes, and plans for dataset release.
- In response, the authors added quantitative evaluations and a user study, provided explicit ablations on adapter initialization (zero-initialization versus copy-initialization) and the number of MPLI planes, discussed generalization through architectural constraints and limited trainable parameters, and confirmed plans to release both the dataset and the model.

---

### Reviewer 53nu (Score: 6)

- The reviewer raised concerns about the limited lighting model (isotropic point lights only), missing quantitative and temporal consistency evaluations, questions about MPLI design choices (such as the number of planes and latent-space alignment), and the fairness of comparisons to text-based baselines.
- In the rebuttal, the authors added quantitative metrics and a user study, provided ablations justifying the choice of the number of MPLI planes, clarified the temporal granularity at which MPLI operates, acknowledged the limitations of the current lighting model, and discussed the inherent limitations of comparisons to text-based methods.

---

### Reviewer TeRx (Score: 6)

- The reviewer raised concerns regarding reproducibility due to reliance on a private T2V backbone, the potentially awkward or unfair comparison setting when evaluating against text-driven methods, insufficient ablations to disentangle the contributions of MPLI versus the backbone and dataset, and the need for broader baseline comparisons.
- In the rebuttal, the authors committed to adapting the method to an open-source video model and releasing data and weights, clarified why existing baselines cannot directly consume explicit lighting inputs, expanded the discussion of related work, and provided additional clarifications regarding design choices.

**Reviewer Scores:**

### Reviewer adZs

- **Original score:** 4
- **Predicted final score:** 4–6
- **Rationale:** The rebuttal responds directly to the reviewer’s main concerns by adding quantitative evaluations, a user study, and additional ablations. However, the concern about the limited lighting representation may not be fully addressed, as the method still only supports point lights (albeit multiple ones). As a result, while a score increase is possible, it is not guaranteed.

---

### Reviewer zrp3

- **Original score:** 4
- **Predicted final score:** 4–6
- **Rationale:** Most of the reviewer’s requests—including quantitative results, user studies, ablations, and clarification of design choices—are addressed to some extent in the rebuttal. However, the concern about limited baseline comparisons may persist, as the additional results still primarily compare against the same two baselines. Therefore, a moderate score increase is possible but not certain.

---

### Reviewer 53nu

- **Original score:** 6
- **Predicted final score:** 6
- **Rationale:** The rebuttal addresses several empirical concerns and clarifies design decisions. Nonetheless, some conceptual limitations—such as the restricted lighting model and questions about baseline fairness—are acknowledged rather than fully resolved. The reviewer is therefore likely to maintain their score.

---

### Reviewer TeRx

- **Original score:** 6
- **Predicted final score:** 6-8
- **Rationale:** The rebuttal mitigates the reviewer’s reproducibility concerns through commitments to open-sourcing and adapting to an open-source backbone. However, some skepticism regarding comparisons to text-driven methods may persist, making a score change possible but less likely.

---

### Decision · Program_Chairs · 2026-01-26

Reject